## Replications

cognition/evolution/psychology

pride, shame, emotion, cooperation, replication, reputation

**Author for correspondence:**
Adam Scott Cohen
e-mail: cohen9@hawaii.edu

# Do pride and shame track the evaluative psychology of audiences? Preregistered replications of Sznycer *et al.* (2016, 2017)

## Adam Scott Cohen[1,2], Rie Chun[1] and Daniel Sznycer[3]

[1]University of Hawai'i at Manoa, Honolulu, HI, USA
[2]Hawaii State Judiciary, Honolulu, HI, USA
[3]University of Montreal, Montreal, Quebec, Canada

(iD) ASC, 0000-0003-2719-3351

Are pride and shame adaptations for promoting the benefits of being valued and limiting the costs of being devalued, respectively? Recent findings indicate that the intensities of anticipatory pride and shame regarding various potential acts and traits track the degree to which fellow community members value or disvalue those acts and traits. Thus, it is possible that pride and shame are engineered to activate in proportion to others' valuations. Here, we report the results of two preregistered replications of the original pride and shame reports (Sznycer *et al.* 2016 *Proc. Natl Acad. Sci. USA* **113**, 2625–2630. (doi:10.1073/pnas.1514699113); Sznycer *et al.* 2017 *Proc. Natl Acad. Sci. USA* **114**, 1874–1879. (doi:10.1073/pnas.1614389114)). We required the data to meet three criteria, including frequentist and Bayesian replication measures. Both replications met the three criteria. This new evidence invites a shifting of prior assumptions about pride and shame: these emotions are engineered to gain the benefits of being valued and avoid the costs of being devalued.

## 1. Introduction

Humans have a powerful need to belong [1–3] and, reciprocally, a powerful aversion to being devalued, excluded or ostracized [4–6]. Here, we consider a functional framework to test emotional components of this motivational disposition. More specifically, we test the emotions of pride and shame as systems engineered to promote others' positive valuations of the self and to avoid being devalued by others [7–12].

Humans are a social species. The ancestral social ecology of humans featured a host of ecological challenges: food scarcity and variability [13], non-human predators, aggressive conspecifics [14] and serious bouts of incapacitation due to injury and disease [15]. Modern technologies and institutions that minimize those dangers were absent. Nevertheless, those challenges could be buffered, to some extent, to the extent that other individuals in one's group were willing and able to render aid. One possibility then is that natural selection operating on these features of the social ecology of ancestral humans could have selected for regulatory adaptations targeting the social-evaluative psychology in the brains of other individuals for the purpose of (i) inducing others to augment and maintain their positive social evaluations of the self and (ii) avoiding the likelihood and costs of being socially devalued by others. This is believed to be the context in which the human emotions of pride and shame evolved [8–12].

Pride, for instance, serves multiple interrelated functions. It impels the individual to (i) perform actions or develop personal characteristics that other people value, (ii) transmit information about achievements so that others can register the new state of affairs, and (iii) take advantage of the increased valuation from others by e.g. demanding better treatment [8,11,12,16,17].

Many features of pride documented in the emotion literature can be understood by reference to these hypothesized functions. For example, pride is among the most positively valenced emotions [18]. This hedonic feature may serve as incentive to continue performing the actions and developing the characteristics that would bring about positive evaluations from others [17]. Pride has a prototypical display of achievement that includes expansion of body size, a zoologically ubiquitous index of dominance and physical formidability [8,12,19,20]. Research has shown that observers interpret the human pride display as indicating dominance and achievement [21]. Moreover, children [22] and adults around the world recognize the human pride display [23].

Shame, like pride, can be understood as a system that aims to operate on others' internal representations of oneself and the degree to which one is socially valued. Shame has the complementary function of minimizing the spread of adverse information about the self and the reputational damage that occurs when such information reaches the minds of others [8,9,12,24].

Further, the design features of shame, like the design features of pride, can be understood in the light of the corresponding hypothesized functions. For instance, when information signals that others might learn about actions taken by the self that benefit the self but harm others, shame can terminate the execution of those actions [25,26]. Further, when people feel shame, they hide and suppress incriminating evidence [27–29]. Shame has a prototypical full-body display, the antithesis of the pride display, which conveys submission or appeasement vis-à-vis the devaluing audience [8,12,30]. This voluntary conveyance of acceptance of reduced valuation from others can be understood as 'making the best of a bad situation', as evidence indicates that audiences have more negative evaluative reactions when, for instance, a wrongdoer does not produce a shame display [31].

Although pride and shame have been studied extensively, dissecting these emotions from the standpoint of their hypothesized target domain—the evolved social-evaluative psychology of audiences—is comparatively infrequent. This is unfortunate, because over evolutionary time the evaluative psychology of audiences dictated (i) the courses of action an individual was to adopt if others were to value her, and (ii) the information-processing structure of the pride and shame systems tasked with gaining valuation and avoiding devaluation. Thus, the evaluative psychology of audiences is key to mapping the cognitive architecture of pride and shame. Indeed, recent reports have demonstrated close quantitative correspondences between the activation of pride and shame, on the one hand, and the direction and magnitude of audience's social evaluations, on the other hand.

## 1.1. Prior evidence that pride and shame may track the valuations of audiences

A well-engineered pride system must mobilize not only reactively but also prospectively, in order to motivate the pursuit of socially valued actions that might increase others' valuations of the self [16,32]. In this way, prospective pride helps the individual decide which courses of action to take.

It has been hypothesized that the anticipatory feeling of pride is an internally generated prediction that signals the magnitude of audience valuation one would accrue if one took an action that others value [16]. A pride system that accurately forecasts and precisely tracks audience valuation allows the individual to avoid two types of costly errors: (i) *under-activation of anticipatory pride*, which would cause the individual to insufficiently pursue socially valued courses of action, and (ii) *over-activation of anticipatory pride*, which would cause the individual to over-pursue actions in excess of their actual return. This analysis suggests the existence of a feature: the pride system should (i) forecast the

magnitude of valuation people in one's social ecology would express if one took a given act that they favour, and (ii) deliver an internal signal (anticipatory pride) whose intensity is proportional to it.

Experiments conducted in 16 countries supported this prediction: the intensity of anticipatory pride in every country closely tracked the magnitude of valuation expressed by local audiences—in the absence of any communication between participants reporting their pride versus audiences reporting their valuation regarding each of various potential acts and traits, such as generosity, trustworthiness and skills [16].

Analogous reasoning suggests that the anticipatory feeling of shame is an internal prediction of the degree to which local audiences would devalue the individual if she took an action that they disfavour, such as theft, sexual infidelity or stinginess ([24]; see [33–35]). By forecasting and tracking the precise magnitude of audience devaluation, the aversive signal of anticipated shame allows the individual to steer adaptively between a dangerous disregard of others' views, which would yield excessive devaluation, and an excessive timidity about one's possible disgraceful behaviour, which would yield insufficient personal payoffs. As predicted, shame closely tracked audience devaluation in three countries [24].

## 1.2. The present work

Here, we present the results of two preregistered replications addressing the following questions: does anticipatory pride track the magnitude of audience valuation [16]? And, does anticipatory shame track the magnitude of audience devaluation [24]?

The present work addresses two issues surrounding the replicability of the original pride and shame studies. First, here we perform preregistered confirmatory analyses. This allows us to validly conduct null hypothesis significance testing while controlling long-run error rates that otherwise would be inflated by undisclosed flexibility in data analysis [36,37]. If the replications are successful, that would make it less likely that the original findings were false positives or that the original effect sizes were misestimated due to undisclosed flexibility in data analysis. Second, the original and replication studies were conducted and analysed by different individuals. Therefore, a successful replication would reduce uncertainty in the original findings by arguing against experimenter error in the original study design, implementation or analysis.

Following best practices to design and implement replication studies [38,39], the first two authors collaborated with the third author (the lead author of the original studies). The first two authors conducted the studies and analysed the data, while the third author shared original materials and data, provided feedback about the accuracy of the study implementation and helped identify discrepancies with the original studies. This allowed us to implement replication studies that were closely aligned with the original studies and document any remaining differences.

The procedures were the same as in the study by Sznycer *et al.* [16,24], with the following exceptions: (i) The original studies included a number of measures/stimuli testing other hypotheses, which were dropped from the replications, (ii) the replications were administered online, as the original studies were, but were conducted exclusively in laboratory rather than a mix of inside and outside of laboratory, and (iii) participants were students completing the tasks as part of a course assignment rather than paid participants, as was the case in some of the original samples. Successful replications would suggest the original effects are robust to these modified procedures. Although there were not strong theoretical reasons for expecting these procedural differences to alter the original effects, these differences were preregistered, as they would be among the first factors to consider as moderators in explaining any failures to replicate.

The current studies also differed from the original research by using the same participants across studies. This difference could impact the results even though study order and condition assignment were random. If participants were first assigned to the audience devaluation condition in the Shame study and then received the pride condition in the other study, it is possible that those participants continued to adopt an audience perspective in the pride condition despite instructions to anticipate how much pride they would feel. Adopting an audience perspective in the pride condition would inflate correlations between pride and audience valuations. The same concern would apply to other condition pairs in which perspective shifts between the first and second conditions (e.g. receiving valuation before shame, pride before devaluation and shame before valuation). This potential problem was addressed with *post hoc* analyses that excluded data from the study that was administered second, eliminating the possibility of an audience devaluation (or valuation) condition priming an audience perspective in a subsequent pride (or shame) condition and inflating the correlations.

Following Brandt *et al.*'s template [38], we preregistered hypotheses, *a priori* power analyses,[1] and data analysis plans on the open science framework for Study 1 (https://osf.io/uyfqw) and Study 2 (https://osf.io/gvmun).

Since no individual measure of replication success is without limitation [40–42], we defined replication success as meeting three criteria: (i) a correlation between emotion and audience evaluations that is statistically significant ($p < 0.05$) and in the same direction as in the original study, (ii) an effect size that is different from zero and not different from the original effect size, and (iii) a replication Bayes factor [43] that exceeds 3,[2] which is considered at a minimum 'substantial evidence' in favour of the alternative hypothesis relative to the null hypothesis [44].

# 2. Study 1—pride

## 2.1. Method

### 2.1.1. Participants

We recruited 87 participants. Six were removed from analyses for failing an attention check, leaving a final sample size of 81 participants ($M = 20.8$ years, s.d. = 4.55 and 57 females). Participants were students from the University of Hawai'i at Mānoa who were enrolled in a research methods course [45]. Although students worked with the data as part of a course project after participating in the study, they were naive to hypotheses at the time of testing. Bootstrapping simulations on the Study 1 data from the US sample of [16] indicated that 95% power would be achieved with 10 participants. However, because participation was part of a course requirement, the stopping rule[3] for the frequentist tests (Replication Criteria i and ii) was determined by the number of students enrolled in the course ($n = 87$). The actual sample, after exclusion criteria were applied ($n = 81$), was well in excess of the sample size that would produce 95% power.

### 2.1.2. Design

Study 1 tested whether the anticipated intensity of felt pride with respect to a given prospective act or trait that others positively value correlates with the degree of positive valuation attached to that act or trait by those in the social world of the individual. Participants rated 25 brief hypothetical scenarios in which someone's acts or traits might cause them to be viewed positively by others.

Participants were randomly assigned to an *audience* condition or a *pride* condition. In the audience condition, participants were asked to rate 25 scenarios involving another individual (e.g. 'Her children are healthier and taller than average for their age', 'She is ambitious'). Participants in the audience condition were asked to 'indicate how you would view [someone of your same sex and age] if they were in those situations,' on scales ranging from 1 (I wouldn't view her positively at all) to 7 (I'd view her very positively). These ratings provide event-specific measures of positive social valuation.

In the pride condition, a different set of participants were asked to 'indicate how much pride you would feel if you were in those situations' (i.e. in the 25 scenarios; e.g. 'Your children are healthier and taller than average for their age', 'You are ambitious'), on scales ranging from 1 (no pride at all) to 7 (a lot of pride). Except for the perspectival differences, the stimuli in the pride and audience conditions were identical. The scenarios were presented in randomized order in both conditions. To conduct the replications, we used original materials provided by the third author.

### 2.1.3. Procedure

Participants were tested in a computer laboratory. They participated in Study 1, Study 2 and a third unrelated study in random order. Participants entered their gender and age, were randomly assigned to one of the two conditions, and completed the task. The scenarios were gendered according to the participant's gender. Participants were given an attention check before completing the study.

---

[1]We report a different power analysis in the Method sections of Studies 1 and 2 to address a problem with the preregistered power analyses. See the addendum on OSF for details: https://osf.io/jymzk/.

[2]The Bayes factor quantifies the evidence in favour of one hypothesis relative to a second hypothesis. A Bayes factor of 3 represents odds of 3:1 in support of a hypothesis compared with another, competing hypothesis.

[3]Due to an oversight, the stopping rules for Studies 1 and 2 were not preregistered. However, no analyses were conducted before the stopping rule was reached and no additional data were collected after.

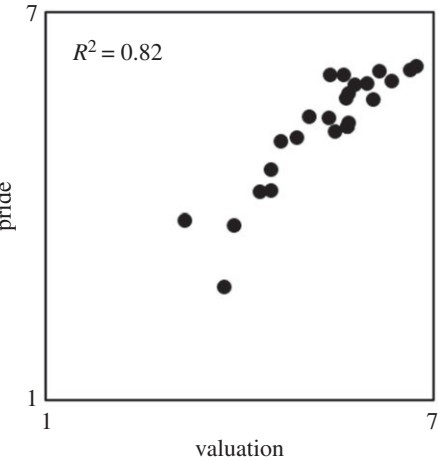

**Figure 1.** Scatter plot: pride as a function of valuation. Each point represents the mean valuation rating and mean pride rating of one scenario. Valuation and pride ratings were given by different participants. $N$ = number of scenarios = 25. Effect size: $R^2$ linear.

## 2.2. Results

This article received results-blind in-principle acceptance (IPA). Following IPA, the accepted Stage 1 version of the manuscript, not including results and discussion, was preregistered on the OSF (https://osf.io/8r9ah). This preregistration was performed after data analysis. Materials, data and analyses are available on the OSF (https://osf.io/upg5w/).

### 2.2.1. Is the correlation between pride and audience valuation significantly different from zero and in the same direction as in the original study?

Yes. For each scenario, we calculated the mean pride ratings provided by participants in the pride condition, and the mean valuation ratings provided by participants in the audience condition. As in the original study, the pride means and the valuation means were positively correlated, $r_{23} = 0.90$, $p < 0.001$, 95% CI [0.79, 0.96] (figure 1; Mean pride and valuation ratings are displayed in table 1). Recall that the pride and valuation ratings originate from different participants. Consequently, this high correlation cannot be attributed to participants matching their pride and valuation ratings.

### 2.2.2. Is the effect size different from zero and not different from the original effect size?

Yes. Using $r$ as a measure of effect size, the 95% confidence interval from the replication study, 95% CI [0.79, 0.96], was not consistent with an effect size of 0. Second, the $r$ from the replication study, $r_{23} = 0.90$, fell within the 95% confidence interval from the original study, 95% CI [0.69, 0.93].

### 2.2.3. Is the replication Bayes factors greater than 3 and in favour of the alternative hypothesis relative to the null hypothesis?

Yes. We complemented the previous two frequentist criteria by computing an evidence updating Bayes factor (EU-BF). The EU-BF indicates whether an effect is present in a replication experiment factoring in the data from the original experiment [43]. The EU-BF is calculated as follows: $BF_{10}$ (replication | original) = $BF_{10}$ (original + replication)/$BF_{10}$ (original). We took two approaches to calculate a valid Bayesian measure of replication success (E.J. Wagenmakers and Alexander Ly, personal communication).[4] First, we combined the 25 item pairs from the original and the replication studies as if they were separate items, yielding 50 item pairs, to calculate $BF_{10}$ (original + replication). Dividing this by $BF_{10}$ (original) yielded $BF_{10}$ (replication | original) = $1.03 \times 10^9$, which exceeded a Bayes factor of 3. For the second approach, we generated a posterior probability for $\rho$ from the original study and used it as the prior for the replication study. The recalculated $BF_{10}$ (replication | original) was $1.19 \times 10^8$, which also exceeded a Bayes factor of 3. The two measures are off by almost a factor of 9 because the second approach

---

[4]An initial calculation of the replication Bayes factor that was invalid is described in the electronic supplementary material.

**Table 1.** Ratings of valuation and pride, by scenario. Note: displayed are means, with standard deviations in parentheses. *N*s: valuation: 42; pride: 39. The male versions of the pride and valuation scenarios are presented before and after the slash, respectively. The female versions of the scenarios read 'husband' (scenario # 20), 'man' (scenario # 20) and 'woman' (scenario # 25), instead of 'wife', 'woman' and 'man'. The female version of the pride scenario # 23 read 'her' instead of 'him'. The female versions of the valuation scenarios featured a female target, so the pronouns were female pronouns. Otherwise, the male and female scenarios were identical. Scenarios are displayed from the highest to the lowest mean valuation scores.

| # | scenario | valuation | pride |
|---|---|---|---|
| 18 | you are trustworthy/he is trustworthy | 6.74 (0.54) | 6.15 (1.25) |
| 4 | you take very good care of your children/he takes very good care of his children | 6.64 (0.76) | 6.10 (1.25) |
| 14 | you are generous with others/he is generous with others | 6.36 (1.03) | 5.92 (1.29) |
| 5 | you can support your children economically/he can support his children economically | 6.17 (0.99) | 6.08 (1.33) |
| 15 | you are ambitious/he is ambitious | 6.07 (1.09) | 5.64 (1.39) |
| 22 | people love your sense of humour/people love his sense of humour | 5.98 (1.24) | 5.90 (1.31) |
| 8 | you are very smart/he is very smart | 5.79 (1.18) | 5.87 (1.47) |
| 10 | you have good table manners/he has good table manners | 5.69 (1.26) | 5.28 (1.85) |
| 16 | you have many unique skills/he has many unique skills | 5.69 (1.37) | 5.74 (1.27) |
| 19 | when there is a conflict in your community, people ask you to mediate between the two sides/when there is a conflict in his community, people ask him to mediate between the two sides | 5.67 (1.30) | 5.23 (1.46) |
| 25 | people think that you are the bravest man of your community/people think that he is the bravest man of his community | 5.67 (1.32) | 5.23 (1.75) |
| 2 | you host your extended family for a holiday meal; they think it's the best meal they've ever had/he hosts his extended family for a holiday meal; they think it's the best meal they've ever had | 5.64 (1.25) | 5.67 (1.38) |
| 17 | you have a lot of good friends/he has a lot of good friends | 5.62 (1.32) | 6.03 (1.22) |
| 9 | you have more years of education than those around you/he has more years of education than those around him | 5.48 (1.29) | 5.15 (1.68) |
| 24 | you finished first in a marathon/he finished first in a marathon | 5.40 (1.42) | 6.03 (1.27) |
| 6 | you are playing a throwing game with your friends. All your throws hit the target/he is playing a throwing game with his friends. All his throws hit the target | 5.38 (1.21) | 5.36 (1.66) |
| 3 | your children are healthier and taller than average for their age/his children are healthier and taller than average for their age | 5.07 (1.24) | 5.38 (1.44) |
| 12 | you are physically attractive/he is physically attractive | 4.88 (1.38) | 5.05 (1.67) |
| 20 | your wife is the most attractive woman of your community/his wife is the most attractive woman of his community | 4.64 (1.45) | 5.00 (1.81) |
| 1 | you look ten years younger than you are/he looks ten years younger than he is | 4.48 (1.31) | 4.23 (1.74) |
| 13 | you are wealthy/he is wealthy | 4.48 (1.25) | 4.56 (1.87) |
| 7 | you come from a wealthy family with high status and many connections/he comes from a wealthy family with high status and many connections | 4.31 (1.30) | 4.21 (1.70) |
| 23 | an acquaintance of yours had been bullying you for a while. At some point you got tired of it, and you beat him very badly. You were never bullied again/an acquaintance of his had been bullying him for a while. At some point he got tired of it, and he beat him very badly. He was never bullied again. | 3.90 (2.00) | 3.69 (1.92) |

(*Continued.*)

| # | scenario | valuation | pride |
|---|----------|-----------|-------|
| 21 | you are the tallest in your circle of friends/he is the tallest in his circle of friends | 3.76 (1.54) | 2.74 (1.45) |
| 11 | you get into a fight in front of everybody and you completely dominate your opponent with punch after punch, until your opponent is knocked out/he gets into a fight in front of everybody and he completely dominates his opponent with punch after punch, until his opponent is knocked out | 3.14 (2.03) | 3.77 (1.99) |

approximates the first using a stretched β-distribution, which is less accurate for correlations towards the tails of the distribution (e.g. near $\rho = 1$), as in the current case. Both replication Bayes factors provide evidence consistent with the frequentist approaches, satisfying the third criterion for a successful replication.

# 3. Study 2—shame

## 3.1. Method

### 3.1.1. Participants

The sample for Study 2 was the same as Study 1. Participants were excluded from analyses if they failed the attention check from Study 1.[5] Bootstrapping simulations on the Study 1 data from the US sample of Sznycer *et al.* [24] indicated that 95% power would be achieved with 14 participants. However, because participation was part of a course requirement, the stopping rule for the frequentist tests (Replication Criteria i and ii) was determined by the number of students enrolled in the course ($n = 87$). Therefore, the actual sample ($n = 81$ after removing inattentives) well exceeded the sample size that would produce 95% power.

### 3.1.2. Design

Study 2 tested whether the anticipated intensity of felt shame with respect to a prospective act or trait that others disvalue tracks the degree of devaluation expressed by local audiences regarding that act or trait. Participants rated 29 brief hypothetical scenarios in which someone's acts or traits might lead them to be viewed negatively. The scenarios featured situations in various evolutionarily relevant domains, including social exchange, parenting, mating, the violation of social norms, status and skills.

Participants were randomly assigned to an *audience* condition or a *shame* condition. In the audience condition, participants were asked to rate 29 scenarios involving another individual (e.g. 'He hosts his extended family for a holiday meal, but he burns the food', 'He dropped out of school much earlier than others'). In this condition, participants were asked to 'indicate how you would view this person if they were in those situations'; they indicated their reactions using scales ranging from 1 (I wouldn't view him negatively at all) to 7 (I'd view him very negatively). These ratings provide event-specific measures of the degree to which the members of a given population would devalue the individual described in the scenarios.

A different set of participants were asked, in the shame condition, to 'indicate how much shame you would feel if you were in those situations' (i.e. in each of the 29 scenarios; e.g. 'You host your extended family for a holiday meal, but you burn the food', 'You dropped out of school much earlier than others'), on 1 (no shame at all) to 7 (a lot of shame) scales. In both conditions, the scenarios were presented in a randomized order. To conduct the replications, we used original materials provided by the third author.

### 3.1.3. Procedure

The procedures were the same as in Study 1, except here participants were randomly assigned to an audience condition or a shame condition, and no attention check was administered.

---

[5]An attention check was used in the original Shame study [24] but was not provided with materials for Study 2, so the session was run without a second attention check.

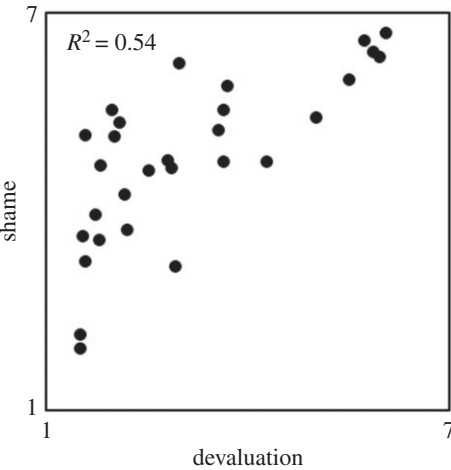

$R^2 = 0.54$

shame

devaluation

**Figure 2.** Scatter plot: shame as a function of devaluation. Each point represents the mean devaluation rating and mean shame rating of one scenario. Devaluation and shame ratings were given by different participants. $N$ = number of scenarios = 29. Effect size: $R^2$ linear.

## 3.2. Results

### 3.2.1. Is the correlation between shame and devaluation significantly different from zero and in the same direction as in the original study?

Yes. For each scenario, we calculated the mean shame ratings provided by participants in the shame condition, and the mean devaluation ratings provided by participants in the audience condition. The shame means and the devaluation means were positively correlated, $r_{27} = 0.73$, $p < 0.001$ and 95% CI [0.50, 0.87] (figure 2; Mean shame and devaluation ratings are displayed in table 2).

### 3.2.2. Is the effect size different from zero and not different from the original effect size?

Yes. Using $r$ as a measure of effect size, the 95% confidence interval from the replication study, 95% CI [0.50, 0.87], was not consistent with an effect size of zero. Second, the $r$ from the replication study, $r_{27} = 0.73$, fell within the 95% confidence interval from the original study, 95% CI [0.48, 0.86].

### 3.2.3. Is the replication Bayes factors greater than 3 and in favour of the alternative hypothesis relative to the null hypothesis?

Yes. We used the same two Bayesian approaches from Study 1. On the first approach, we combined the *29 item pairs from the original and replication studies* as if they were separate items (total = 58 item pairs) to calculate $BF_{10}$ (original + replication). Dividing this by $BF_{10}$ (original) yielded $BF_{10}$ (replication | original) = $2.76 \times 10^4$, which exceeded a Bayes factor of 3. On the second approach, we generated a posterior for $\rho$ from the original study and used it as the prior for the replication study. The recalculated $BF_{10}$ (replication | original) was $4.55 \times 10^4$, which also exceeded a Bayes factor of 3. Both replication Bayes factors are consistent with the frequentist replication analyses, and together provide evidence for replication of the original results.

## 4. Discussion

Two preregistered replications provided confirmatory evidence in support of evolutionary–functional theories of shame and pride, as reported originally by Sznycer *et al*. ([16,24]; see also [46]). The intensity of anticipatory pride felt with respect to a prospective action or trait closely tracks the magnitude of positive valuation audiences express with respect to that action or trait. Similarly, anticipatory shame closely tracks the audience devaluation.

There is no single, definitive measure of replication success, so we required the data to meet three different criteria to classify the replication as successful: (i) significant replication $p$-values, with effects in the same direction as in the original studies, (ii) replication effect sizes different from zero and

**Table 2.** Ratings of devaluation and shame, by scenario. Note: displayed are means, with standard deviations in parentheses. *N*s: shame: 42, devaluation: 39. The male versions of the shame and devaluation scenarios are presented before and after the slash, respectively. The female versions of the scenarios read 'men' (scenario # 1) and 'husband' (scenarios # 3, 6, 10, 11 and 23) instead of 'women' and 'wife'. Further, the female versions of the devaluation scenarios featured female pronouns. Otherwise, the male and female scenarios were identical. Scenarios are displayed from the highest to the lowest mean devaluation scores.

| # | scenario | devaluation | shame |
|---|---|---|---|
| 3 | at the wedding of an acquaintance, you are discovered cheating on your wife with a food server/at the wedding of an acquaintance, he is discovered cheating on his wife with a food server | 6.05 (1.65) | 6.69 (0.90) |
| 21 | you stole goods from a shop owned by your neighbour/he stole goods from a shop owned by his neighbour. | 5.97 (1.29) | 6.33 (0.98) |
| 10 | everyone discovers that you are sexually unfaithful to your wife/everyone discovers that he is sexually unfaithful to his wife | 5.87 (1.32) | 6.40 (1.15) |
| 7 | you do a bad job taking care of your children/he does a bad job taking care of his children | 5.74 (1.31) | 6.57 (0.83) |
| 20 | you stole goods from a shop owned by a foreign merchant/he stole goods from a shop owned by a foreign merchant | 5.51 (1.48) | 5.98 (1.42) |
| 28 | you are not generous with others/he is not generous with others | 5.03 (1.39) | 5.40 (1.62) |
| 19 | you have poor table manners/he has poor table manners | 4.28 (1.64) | 4.74 (1.81) |
| 24 | you get into a fight in front of everybody and your opponent completely dominates you with punch after punch until you're knocked out/he gets into a fight in front of everybody and his opponent completely dominates him with punch after punch until he's knocked out | 3.69 (1.94) | 5.88 (1.48) |
| 23 | an acquaintance is inappropriately flirting with your wife in front of everybody. Because you're too scared to pick a fight with your rival you remain silent without doing or saying anything/an acquaintance is inappropriately flirting with his wife in front of everybody. Because he's too scared to pick a fight with his rival he remains silent without doing or saying anything. | 3.64 (2.06) | 5.52 (1.63) |
| 29 | you are not very ambitious/he is not very ambitious | 3.64 (1.69) | 4.74 (1.65) |
| 18 | you dropped out of school much earlier than others/he dropped out of school much earlier than others | 3.56 (1.83) | 5.21 (1.92) |
| 8 | you cannot support your children economically/he cannot support his children economically | 2.97 (1.60) | 6.24 (1.30) |
| 1 | you are single. You have a promiscuous sexual life with women/he is single. He has a promiscuous sexual life with women | 2.92 (1.95) | 3.17 (2.05) |
| 15 | your brother stole money from a stranger. As soon as you found out about that, you reported him to the police. How much shame would you feel about your reporting your brother to the police?/his brother stole money from a stranger. As soon as he found out about that, he reported him to the police. How much shame would he feel about his reporting his brother to the police? | 2.87 (1.79) | 4.64 (2.09) |
| 17 | your father defrauded a foreign company/his father defrauded a foreign company | 2.82 (1.65) | 4.76 (1.92) |
| 16 | you are not very smart/he is not very smart | 2.54 (1.45) | 4.62 (1.90) |

(*Continued.*)

| # | scenario | devaluation | shame |
|---|---|---|---|
| 4 | you look ten years older than you are/he looks ten years older than he is | 2.21 (1.36) | 3.71 (1.98) |
| 5 | you host your extended family for a holiday meal, but you burn the food/ he hosts his extended family for a holiday meal, but he burns the food | 2.18 (1.37) | 4.26 (1.82) |
| 14 | your brother stole money from a stranger. How much shame would you feel about your brother stealing money from the stranger?/his brother stole money from a stranger. How much shame would he feel about his brother stealing money from the stranger? | 2.1 (1.55) | 5.33 (1.63) |
| 25 | you are performing a ceremony in front of your community. In the middle of it, your mind goes blank and you forget what to do next/he is performing a ceremony in front of his community. In the middle of it, his mind goes blank and he forgets what to do next | 2.03 (1.22) | 5.12 (1.81) |
| 11 | everyone discovers that your wife is sexually unfaithful to you/everyone discovers that his wife is sexually unfaithful to him. | 1.97 (1.53) | 5.52 (1.92) |
| 26 | you are not physically attractive/he is not physically attractive | 1.82 (1.17) | 4.69 (1.77) |
| 12 | you are playing a throwing game with your friends. All your throws miss the target by a wide margin/he is playing a throwing game with his friends. All his throws miss the target by a wide margin | 1.79 (1.13) | 3.57 (1.88) |
| 9 | you receive welfare money from the government because you cannot financially support your family/he receives welfare money from the government because he cannot financially support his family | 1.74 (1.14) | 3.95 (2.06) |
| 2 | you were in an accident and your face was permanently disfigured/he was in an accident and his face was permanently disfigured. | 1.59 (0.91) | 5.14 (1.80) |
| 13 | you come from a very poor family with low status and no connections/he comes from a very poor family with low status and no connections | 1.59 (0.99) | 3.24 (1.81) |
| 27 | you are poor/he is poor | 1.54 (0.85) | 3.62 (1.83) |
| 6 | your wife makes more money than you do/his wife makes more money than he does | 1.51 (1.23) | 2.14 (1.62) |
| 22 | you have no idea how to load or fire a gun/he has no idea how to load or fire a gun | 1.51 (1.19) | 1.93 (1.50) |

within the 95% confidence interval of the original effect sizes, and (iii) replication Bayes factors, conditioned on the original results, supporting the presence of a correlation, compared with the hypothesis of no-correlation. In both studies, all three criteria of replication success were met. Because the data analysis plans were preregistered and neither the last author nor any of the other authors involved in the original studies participated in the data collection or analyses of the present replications, the original findings are unlikely to be false positives due to researcher degrees of freedom in data collection or analysis [47] or experimenter error.

A recent large-scale replication study showed that replication success largely does not depend on the study sample [48]. Consistent with this, differences between the original and replication samples did not impact the results. Even though the replication sample was drawn from the sixth most ethnically diverse campus in the US [49], probably making it less homogeneous than most of the samples from the original studies, both pride and shame effects replicated.

Although the evidence for replication is strong, there are potential limitations that remain unaddressed. First, in the pride and shame conditions, there were probably multiple scenarios that were not relatable to most participants (e.g. 'You finished first in a marathon' in Study 1; 'You were in an accident and your face was permanently disfigured' in Study 2). How does an individual generate

a specific rating of anticipatory pride (or shame) regarding a situation she has never directly encountered? One possibility is that the individual generates those ratings by imagining how she (or others) would view someone else in that situation. But, if so, that would be equivalent to the task in the audience condition, and a high correlation between audience evaluations and 'pride' (or 'shame') ratings would be inevitable. Under this hypothesis, the ostensible pride ratings in fact reflect ratings of valuation, so the finding is only apparently about pride tracking valuation but in reality is about consensus in valuation. And, similarly, the shame-devaluation findings might simply reflect consensus in shame. It is possible, then, that the observed correlations are unduly inflated by uncommon scenarios that may induce an audience perspective in the pride or shame conditions. Future studies would alleviate these conceptual validity concerns by attempting to replicate the effects with hypothetical events more relatable to participants or with events actually experienced by participants.

Participants may have treated the shame and pride conditions from an audience perspective for another reason. Participants completed one condition of Study 1, one condition of Study 2, and a third unrelated task, all in a random order. Consider a participant assigned first to the audience (valuation) condition of Study 1 and then to the shame condition of Study 2. It is possible that the initial assignment to the audience condition primed the participant to subsequently adopt an audience perspective when generating the shame ratings.[6] If so, the correlation between those shame ratings and the devaluation ratings (provided by other participants) might have been artificially inflated. To test this priming alternative, we analysed only the data from the replication study that was administered first, excluding from analyses the data from the replication study that was administered subsequently. This eliminated any possibility of one study priming the other. Analyses based only on the data of the study presented first did not change the results (see Priming-Control Analyses in the electronic supplementary material). Thus, priming of the other perspective cannot explain the observed effects.

Second, none of the scenarios in either study were negatively worded. Given this, some types of response biases (e.g. acquiescence) may have the effect of artificially inflating the emotion–evaluation correlations. Future studies can profitably assess whether the links between shame and pride on the one hand and the evaluative psychology of audiences on the other generalize to measures other than self-reports.

Because of their association with undesirable outcomes such as aggression and dominance, it has been argued that shame and the hubristic facet of pride are maladaptive emotions ([50–52]; but see [11]). However, the present results suggest a different view of shame and pride. The fact that anticipatory shame tracks the devaluative psychology of audiences suggests that the shame system is designed to steer precisely between a reckless disregard of others' values and an excessive diffidence in the pursuit of personal payoffs. Notwithstanding shame's association with aggression, this emotion comprises multiple features that appear well designed to counter the threat of being devalued (see above). Moreover, even aggression may be a best response in situations where social benefits are no longer as abundantly provided because one is devalued and instead must be bargained for by threatening harm (see [8,12,27]). Similarly, the fact that anticipatory pride tracks audience valuation suggests that this emotion is naturally selected to avoid the insufficient pursuit of success on the one hand and the excessive pursuit, advertisement and entitlement over one's successes on the other.

Ethics. Research was approved by the Office of Research Compliance Human Studies Program at the University of Hawaii at Manoa. The protocol number is 2018-00435. Informed consent was obtained.

Data accessibility. Data have been deposited in an external repository: https://osf.io/upg5w/.

Authors' contributions. A.S.C. implemented the replication studies and collected data; A.S.C. and R.C. analysed the data and drafted the manuscript; D.S. designed the original studies that were replicated and helped draft the manuscript; A.S.C. and D.S. gave the final approval for preregistration. All authors gave final approval for publication and agree to be held accountable for the work performed therein.

Competing interests. We declare we have no competing interests.

Funding. We received no funding for this study.

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
