## [Reviewer comments · Royal Society Open Science]

Review History

RSOS-191922.R0 (Original submission)

Review form: Reviewer 1

Do you have any ethical concerns with this paper?

No

Have you any concerns about statistical analyses in this paper?

No

Recommendation?

Accept with minor revision

Comments to the Author(s)

Overall I think the proposed methods would be a valid and robust replication. The researchers have noted most of the differences between their study and the original studies in the manuscript and/or in the preregistration.

The one concern I have with the procedure as described both being a valid replication and being a valid test of the hypotheses anchors on the fact that the same individuals were participants in

both Study 1 and Study 2. The original studies were published in two different papers, with two different samples, so I assume in the original studies these were different sets of participants. One of the arguments made in both the original papers is that correlations between the audience condition ratings and the shame/pride ratings provide powerful evidence that each of the emotions tracks the evaluative psychology of the audience precisely because the ratings in each condition came from different people. Based on my reading of the procedure, it would have been possible, for example, to be assigned the audience condition in Study 2, then be assigned to the pride condition in Study 1. It is possible that a participant's ratings would then be influenced by thinking of the audience perspective, or an awareness that the researchers were also assessing the audience perspective, even though they were directed to rate their own emotional response in the next study. This departure from the original studies is concerning because it could generate a positive correlation between the audience and emotion ratings for reasons other than pride and shame tracking evaluative psychology of audiences. At a minimum, this departure from the original study methodologies should be addressed in the manuscript, possibly as a limitation. The analysis plan should also include any plans to address order effects. I did not see this included in the preregistration plan, so if such analyses are or were not planned a priori they need to be clearly described as post hoc.

I had one minor concern with the attention check rule data exclusion rule. From the participant perspective it may have seemed like they were participating in one study (rather than 3) because they completed everything in one session. It sounds like the attention check was embedded in the Study 1 questions, but if data collection for both studies happened in the same session, it would be reasonable to apply the data exclusion rule across both studies- to exclude participant data from both Study 1 and 2 if they failed the one attention check. Both preregistrations state "failure to respond to an attention check" as exclusion criteria. Ideally, excluding those participants from Study 2 won't change any results, but if it does then I would want more justification for why it was appropriate to exclude their data from Study 1 but not Study 2.

Review form: Reviewer 2

Do you have any ethical concerns with this paper?

No

Have you any concerns about statistical analyses in this paper?

No

Recommendation?

Accept in principle

Comments to the Author(s)

See the attached file for my review (Appendix A).

Decision letter (RSOS-191922.R0)

17-Feb-2020

Dear Dr Cohen,

On behalf of the Editors, I am pleased to inform you that your Manuscript RSOS-191922 entitled "Do Pride and Shame Track the Evaluative Psychology of Audiences?: Preregistered Replications

of Sznycer et al. (2016, 2017)" deemed suitable for in-principle acceptance in Royal Society Open Science subject to minor revision in accordance with the referee and editor suggestions. Please find their comments at the end of this email.

The reviewers and handling editors have recommended publication, but also suggest some minor revisions to your manuscript. Therefore, I invite you to respond to the comments and revise your manuscript.

Please you submit the revised version of your manuscript within 7 days (i.e. by the 25-Feb-2020). If you do not think you will be able to meet this date please let me know immediately.

When submitting your revised manuscript, you will be able to respond to the comments made by the referees and upload a file "Response to Referees" in the "File Upload" step. You can use this to document any changes you make to the original manuscript. In order to expedite the processing of the revised manuscript, please be as specific as possible in your response to the referees.

Full author guidelines can be found here <https://royalsocietypublishing.org/rsos/replication-studies#AuthorsGuidance>.

on behalf of Professor Chris Chambers (Registered Reports Editor, Royal Society Open Science)
openscience@royalsociety.org

Associate Editor Comments to Author (Professor Chris Chambers):

Two expert reviewers have now assessed the Stage 1 manuscript. Both reviewers and positive and indicate that the two Stage 1 primary criteria are met pending minor revisions. Reviewer 1 notes a potential deviation from the original methods and asks for clarification of the data exclusion conditions for the attention check. Reviewer 2 is similarly enthusiastic and notes the importance of making available the stimuli and materials (which is requirement at Royal Society Open Science). Provided the authors are able to respond thoroughly to all points, in principle acceptance should be forthcoming without requiring further in-depth Stage 1 review.

Reviewers' comments to Author:

Reviewer: 1

Comments to the Author(s)

Overall I think the proposed methods would be a valid and robust replication. The researchers have noted most of the differences between their study and the original studies in the manuscript and/or in the preregistration.

The one concern I have with the procedure as described both being a valid replication and being a valid test of the hypotheses anchors on the fact that the same individuals were participants in both Study 1 and Study 2. The original studies were published in two different papers, with two different samples, so I assume in the original studies these were different sets of participants. One of the arguments made in both the original papers is that correlations between the audience condition ratings and the shame/pride ratings provide powerful evidence that each of the emotions tracks the evaluative psychology of the audience precisely because the ratings in each condition came from different people. Based on my reading of the procedure, it would have been possible, for example, to be assigned the audience condition in Study 2, then be assigned to the pride condition in Study 1. It is possible that a participant's ratings would then be influenced by thinking of the audience perspective, or an awareness that the researchers were also assessing the audience perspective, even though they were directed to rate their own emotional response in the next study. This departure from the original studies is concerning because it could generate a positive correlation between the audience and emotion ratings for reasons other than pride and shame tracking evaluative psychology of audiences. At a minimum, this departure from the original study methodologies should be addressed in the manuscript, possibly as a limitation. The analysis plan should also include any plans to address order effects. I did not see this included in the preregistration plan, so if such analyses are or were not planned a priori they need to be clearly described as post hoc.

I had one minor concern with the attention check rule data exclusion rule. From the participant perspective it may have seemed like they were participating in one study (rather than 3) because they completed everything in one session. It sounds like the attention check was embedded in the Study 1 questions, but if data collection for both studies happened in the same session, it would be reasonable to apply the data exclusion rule across both studies- to exclude participant data from both Study 1 and 2 if they failed the one attention check. Both preregistrations state "failure to respond to an attention check" as exclusion criteria. Ideally, excluding those participants from Study 2 won't change any results, but if it does then I would want more justification for why it was appropriate to exclude their data from Study 1 but not Study 2.

Reviewer: 2
Comments to the Author(s)

See the attached file for my review.

Author's Response to Decision Letter for (RSOS-191922.R0)

See Appendix B.

Decision letter (RSOS-191922.R1)

26-Feb-2020

Dear Dr Cohen

On behalf of the Editor, I am pleased to inform you that your Manuscript RSOS-191922.R1 entitled "Do Pride and Shame Track the Evaluative Psychology of Audiences?: Preregistered

Replications of Sznycer et al. (2016, 2017)" has been accepted in principle for publication in Royal Society Open Science. The reviewers' and editors' comments are included at the end of this email.

You may now progress to Stage 2 and complete the study as approved.

Please note that you must now register your approved protocol on the Open Science Framework (<https://osf.io/rr>), using the 'Submit your approved Registered Report' option and then the 'Registered Report Protocol Preregistration' option. Please use the Registered Report option even though your article is being accepted as a Stage 1 Replication. Further into the registration process, in the Journal Title field enter 'Royal Society Open Science (Replication article type, Results-Blind track)'. Please note that a time-stamped, independent registration of the protocol is mandatory under journal policy, and manuscripts that do not conform to this requirement cannot be considered at Stage 2. The protocol should be registered unchanged from its current approved state. Please include a URL to the protocol in your Stage 2 manuscript, and because you submitted via the Results-Blind track please note in the manuscript that the pre-registration was performed after data analysis (e.g. 'This article received results-blind in-principle acceptance (IPA) at Royal Society Open Science. Following IPA, the accepted Stage 1 version of the manuscript, not including results and discussion, was preregistered on the OSF (URL). This preregistration was performed after data analysis.')

We invite you to resubmit your paper with results and conclusions for peer review as a Stage 2 Replication. Please note that your manuscript can still be rejected for publication at Stage 2 if the Editors consider any of the following conditions to be met:

- The Introduction and methods deviated from the approved Stage 1 submission (required).
- The authors' conclusions were not considered justified given the data.

We encourage you to read the complete guidelines for authors concerning Stage 2 submissions at: <https://royalsocietypublishing.org/rsos/replication-studies#AuthorsGuidance>. Please especially note the requirements for data sharing and that withdrawing your manuscript will result in publication of a Withdrawn Registration.

Once again, thank you for submitting your manuscript to Royal Society Open Science and I look forward to receiving your Stage 2 submission. If you have any questions at all, please do not hesitate to get in touch.

Kind regards,
Andrew Dunn
Royal Society Open Science
openscience@royalsociety.org

on behalf of Professor Chris Chambers (Registered Reports Editor, Royal Society Open Science)
openscience@royalsociety.org

Author's Response to Decision Letter for (RSOS-191922.R1)

See Appendix C.

RSOS-191922.R2 (Revision)

Review form: Reviewer 1

Is the manuscript scientifically sound in its present form?

Yes

Is the language acceptable?

Yes

Do you have any ethical concerns with this paper?

No

Have you any concerns about statistical analyses in this paper?

No

Recommendation?

Accept with minor revision

Comments to the Author(s)

I have very few comments- I believe the manuscript and the studies meet the criteria needed for publication. The authors addressed the limitations and concerns brought up in Stage 1 and addressed them appropriately. The introduction and methods were generally the same as the original, and any differences were appropriate revisions that did not change what was said in each section.

The three criteria to measure replication success were met. I was not familiar with the EU-BF before this, but based on the author's description and the cited paper it appears to provide another appropriate test for replication, in addition to the other criteria used.

The author's conclusions are justified, and they do a good job acknowledging alternative explanations in the discussion.

A minor edit is needed to Figure 2- I believe the x-axis should display 1-7, like Figure 1, not 1-13.

Review form: Reviewer 2

Is the manuscript scientifically sound in its present form?

Yes

Is the language acceptable?

Yes

Do you have any ethical concerns with this paper?

No

Have you any concerns about statistical analyses in this paper?

No

Recommendation?

Accept as is

Comments to the Author(s)

The authors faithfully fulfilled the studies outlined in Stage 1 of this replication study. The evidence is highly consistent with the original study, and the statistical analyses, both frequentist and bayesian, are clear and easy to understand. The authors also do an excellent job addressing the limitations of the studies in this manuscript in the general discussion and showcase their commitment to best scientific practices and transparency throughout the manuscript and with the supplemental materials and OSF repository. Best wishes to the authors for this and future work!

Decision letter (RSOS-191922.R2)

09-Apr-2020

Dear Dr Cohen

On behalf of the Editor, I am pleased to inform you that your Stage 2 Replication submission RSOS-191922.R2 entitled "Do Pride and Shame Track the Evaluative Psychology of Audiences?: Preregistered Replications of Sznycer et al. (2016, 2017)" has been accepted for publication in Royal Society Open Science subject to minor revision in accordance with the referee suggestions. Please find the referees' comments at the end of this email.

The reviewers and Subject Editor have recommended publication, but also suggest some minor revisions to your manuscript. Therefore, I invite you to respond to the comments and revise your manuscript.

Please also ensure that all the below editorial sections are included where appropriate (a non-exhaustive example is included in an attachment):

- Ethics statement

- Data accessibility

<http://datadryad.org/submit?journalID=RSOS&manu=RSOS-191922.R2>

- Competing interests

- Authors' contributions

- Acknowledgements

- Funding statement

Because the schedule for publication is very tight, it is a condition of publication that you submit the revised version of your manuscript within 7 days (i.e. by the 17-Apr-2020). If you do not think you will be able to meet this date please let me know immediately.

- 1) A text file of the manuscript (tex, txt, rtf, docx or doc), references, tables (including captions) and figure captions. Do not upload a PDF as your "Main Document".
- 2) A separate electronic file of each figure (EPS or print-quality PDF preferred (either format should be produced directly from original creation package), or original software format)
- 3) Included a 100 word media summary of your paper when requested at submission. Please ensure you have entered correct contact details (email, institution and telephone) in your user account
- 4) Included the raw data to support the claims made in your paper. You can either include your data as electronic supplementary material or upload to a repository and include the relevant DOI within your manuscript
- 5) Included your supplementary files in a format you are happy with (no line numbers, Vancouver referencing, track changes removed etc) as these files will NOT be edited in production

Kind regards,
Andrew Dunn
Royal Society Open Science
openscience@royalsociety.org

on behalf of Professor Chris Chambers (Registered Reports Editor, Royal Society Open Science)
openscience@royalsociety.org

Associate Editor Comments to Author (Professor Chris Chambers):

Associate Editor: 1

Comments to the Author:

The Stage 2 manuscript was returned to both of the original Stage 1 reviewers. Happily, both are satisfied that the manuscript meets the Stage 2 criteria. Please attend to the remaining issue identified by Reviewer 1 and full acceptance will follow without further in-depth review.

Reviewers' comments to Author:

Reviewer: 1

Comments to the Author(s)

I have very few comments- I believe the manuscript and the studies meet the criteria needed for publication. The authors addressed the limitations and concerns brought up in Stage 1 and addressed them appropriately. The introduction and methods were generally the same as the original, and any differences were appropriate revisions that did not change what was said in each section.

The three criteria to measure replication success were met. I was not familiar with the EU-BF before this, but based on the author's description and the cited paper it appears to provide another appropriate test for replication, in addition to the other criteria used.

The author's conclusions are justified, and they do a good job acknowledging alternative explanations in the discussion.

A minor edit is needed to Figure 2- I believe the x-axis should display 1-7, like Figure 1, not 1-13.

Reviewer: 2

Comments to the Author(s)

The authors faithfully fulfilled the studies outlined in Stage 1 of this replication study. The evidence is highly consistent with the original study, and the statistical analyses, both frequentist and bayesian, are clear and easy to understand. The authors also do an excellent job addressing the limitations of the studies in this manuscript in the general discussion and showcase their commitment to best scientific practices and transparency throughout the manuscript and with the supplemental materials and OSF repository. Best wishes to the authors for this and future work!

Author's Response to Decision Letter for (RSOS-191922.R2)

See Appendix D.

Decision letter (RSOS-191922.R3)

16-Apr-2020

Dear Dr Cohen:

It is a pleasure to accept your Stage 2 Replication entitled "Do Pride and Shame Track the Evaluative Psychology of Audiences?: Preregistered Replications of Sznycer et al. (2016, 2017)" in its current form for publication in Royal Society Open Science.

on behalf of Professor Chris Chambers (Subject Editor)
openscience@royalsociety.org

Appendix A

Do Pride and Shame Track Evaluative Psychology of Audiences?: Preregistered Replications of Sznycer et al. (2016,2017) Peer Review

This manuscript represents a pivotal piece of conducting valid and reliable research. The authors demonstrate strong commitment to best scientific practices including transparency, pre-registration, large sample sizes, and detailed analysis plans. I address each of the criteria of a Stage 1 review below:

1. *Stage 1 Primary Criterion #1: Whether the authors provide a sufficiently clear and detailed description of the methods for another researcher to closely replicate the proposed experimental procedures and analysis pipeline, and to prevent undisclosed flexibility in the experimental procedures or analysis pipeline.*

The authors' description of the methods is clear and easy to follow. It would be quite simple for another researcher to directly replicate the two experiments described in the manuscript. The analysis plan is clearly detailed in the preregistrations for both studies, but is not mentioned in the methods section of the manuscript. Given the simplicity of the analysis plan (computing correlations), I do not believe it is necessary that this be described in detail at this stage.

The authors clearly define their exclusion criterion and what constitutes a replication in terms of both frequentist and Bayesian criteria. The authors provide samples of situations participants mentally evaluate and each participant evaluates each scenario, so there is little chance of undisclosed flexibility in experimental procedures. If possible, I think readers may benefit from the inclusion of a full list of all the scenarios participants evaluated in the online material.

2. *Stage 1 Primary Criterion #2: Whether the manuscript describes a sufficiently valid (i.e. close) and robust (e.g. statistically powerful) replication of the original study methods and rationale to provide an indication of replicability.*

The manuscript describes a sufficiently valid and robust replication of the original study methods by using the exact same materials as the original study and sample sizes that far surpass what would be required for 95% power based on the original effect size. They clearly define what constitutes a replication of the original effect and point out how this replication complements the original work by pre-registering confirmatory analyses to address the likelihood that original findings were false positives and to more accurately determine the effect size.

3. *Stage 1 Secondary Criterion #1: The logic, rationale, and plausibility of the proposed hypotheses.*

The hypotheses detailed in the manuscript follow directly from previous research on the evolved functions of pride and shame. Much empirical evidence exists supporting the

theoretical perspective from which the authors derive their hypotheses; as such, these hypotheses are a natural extension from prior theory. Given that these hypotheses have already been tested and supported in multiple countries, it is highly plausible that they will be supported again in this nearly identical replication.

4. *Stage 1 Secondary Criterion #2*: The soundness of the methodology and analysis pipeline.

The methodology is sound and sensible, and the chosen analysis pipeline is sufficient for testing these hypotheses.

5. *Stage 1 Secondary Criterion #3*: Whether the authors have considered sufficient outcome-neutral conditions (e.g. absence of floor or ceiling effects; positive controls; other quality checks) for ensuring that the results obtained are able to test the stated hypotheses.

The authors included an attention check in Study 1 to ensure data quality. The participants evaluate a large number of different scenarios for each study, so they will provide a sufficiently wide range of responses that should preclude floor or ceiling effects. That said, providing a complete list of all the stimuli would bolster my confidence that each study contains scenarios that engender ratings of high, medium, and low levels of pride and shame. Additionally, using more items to assess anticipated pride/shame and social value may increase variability in responses if that is an issue.

Overall, I think the authors demonstrated a strong commitment to conducting a faithful replication of the original studies and to transparently planning their work and describing it to readers. My few suggestions are quite minor and none should be considered serious problems.

Appendix B

Associate Editor Comments to Author (Professor Chris Chambers):

Two expert reviewers have now assessed the Stage 1 manuscript. Both reviewers indicate that the two Stage 1 primary criteria are met pending minor revisions. Reviewer 1 notes a potential deviation from the original methods and asks for clarification of the data exclusion conditions for the attention check. Reviewer 2 is similarly enthusiastic and notes the importance of making available the stimuli and materials (which is requirement at Royal Society Open Science). Provided the authors are able to respond thoroughly to all points, in principle acceptance should be forthcoming without requiring further in-depth Stage 1 review.

We appreciate this summary and the opportunity to address the reviewers' comments.

Reviewers' comments to Author:

Reviewer: 1

Comments to the Author(s)

Overall I think the proposed methods would be a valid and robust replication. The researchers have noted most of the differences between their study and the original studies in the manuscript and/or in the preregistration.

The one concern I have with the procedure as described both being a valid replication and being a valid test of the hypotheses anchors on the fact that the same individuals were participants in both Study 1 and Study 2. The original studies were published in two different papers, with two different samples, so I assume in the original studies these were different sets of participants. One of the arguments made in both the original papers is that correlations between the audience condition ratings and the shame/pride ratings provide powerful evidence that each of the emotions tracks the evaluative psychology of the audience precisely because the ratings in each condition came from different people. Based on my reading of the procedure, it would have been possible, for example, to be assigned the audience condition in Study 2, then be assigned to the pride condition in Study 1. It is possible that a participant's ratings would then be influenced by thinking of the audience perspective, or an awareness that the researchers were also assessing the audience perspective, even though they were directed to rate their own emotional response in the next study. This departure from the original studies is concerning because it could generate a positive correlation between the audience and emotion ratings for reasons other than pride and shame tracking evaluative psychology of audiences. At a minimum, this departure from the original study methodologies should be addressed in the manuscript, possibly as a limitation. The analysis plan should also include any plans to address order effects. I did not see this included in the preregistration plan, so if such analyses are or were not planned a priori they need to be clearly described as post hoc.

Thank you for this observation. We have updated the manuscript by noting this overlooked departure from the original study and describing a post hoc analysis to address it:

"The current studies also differed from the original research by using the same participants across studies. This difference could impact the results even though study order and condition assignment were random. If participants assigned to the audience devaluation (or valuation) condition in the Shame (or Pride) study subsequently received the pride (or shame) condition in the other study, and if those participants continued to adopt an audience perspective in the pride (or shame) condition because they had adopted it in the prior study, correlations between pride and audience valuation (or

shame and audience devaluation) would be inflated. This potential problem was addressed with post hoc analyses that excluded data from the study that was administered second, eliminating the possibility of an audience devaluation (or valuation) condition priming an audience perspective in a subsequent pride (or shame) condition and inflating the correlations.” (p. 6-7)

I had one minor concern with the attention check rule data exclusion rule. From the participant perspective it may have seemed like they were participating in one study (rather than 3) because they completed everything in one session. It sounds like the attention check was embedded in the Study 1 questions, but if data collection for both studies happened in the same session, it would be reasonable to apply the data exclusion rule across both studies- to exclude participant data from both Study 1 and 2 if they failed the one attention check. Both preregistrations state “failure to respond to an attention check” as exclusion criteria. Ideally, excluding those participants from Study 2 won’t change any results, but if it does then I would want more justification for why it was appropriate to exclude their data from Study 1 but not Study 2.

The reviewer makes a good point. We have revised the manuscript to indicate that participants will be removed from both studies if they fail the attention check from Study 1. We also updated footnote 4.

“The sample for Study 2 was the same as Study 1. Participants were excluded from analyses if they failed the attention check from Study 1.” (p. 9)

“⁴ An attention check was used in the original Shame study (Sznycer et al., 2016) but was not provided with materials for Study 2, so the session was run without a second attention check.” (p. 9)

Reviewer: 2

Comments to the Author(s)

See the attached file for my review.

Do Pride and Shame Track Evaluative Psychology of Audiences?: Preregistered Replications of Sznycer et al. (2016,2017) Peer Review

This manuscript represents a pivotal piece of conducting valid and reliable research. The authors demonstrate strong commitment to best scientific practices including transparency, pre-registration, large sample sizes, and detailed analysis plans. I address each of the criteria of a Stage 1 review below:

1. *Stage 1 Primary Criterion #1*: Whether the authors provide a sufficiently clear and detailed description of the methods for another researcher to closely replicate the proposed experimental procedures and analysis pipeline, and to prevent undisclosed flexibility in the experimental procedures or analysis pipeline.

The authors’ description of the methods is clear and easy to follow. It would be quite simple for another researcher to directly replicate the two experiments described in the manuscript. The analysis plan is clearly detailed in the preregistrations for both studies, but is not mentioned in the methods section of the manuscript. Given the simplicity of the analysis plan (computing correlations), I do not believe it is necessary that this be described in detail at this stage.

The authors clearly define their exclusion criterion and what constitutes a replication in terms of both frequentist and Bayesian criteria. The authors provide samples of situations participants mentally evaluate and each participant evaluates each scenario, so there is little chance of undisclosed flexibility in experimental procedures. If possible, I think readers may benefit from the inclusion of a full list of all the scenarios participants evaluated in the online material.

We appreciate the reviewer's suggestion and will provide a full list of all scenarios at Stage 2, along with descriptive statistics for each scenario.

2. *Stage 1 Primary Criterion #2*: Whether the manuscript describes a sufficiently valid (i.e. close) and robust (e.g. statistically powerful) replication of the original study methods and rationale to provide an indication of replicability.

The manuscript describes a sufficiently valid and robust replication of the original study methods by using the exact same materials as the original study and sample sizes that far surpass what would be required for 95% power based on the original effect size. They clearly define what constitutes a replication of the original effect and point out how this replication complements the original work by pre-registering confirmatory analyses to address the likelihood that original findings were false positives and to more accurately determine the effect size.

3. *Stage 1 Secondary Criterion #1*: The logic, rationale, and plausibility of the proposed hypotheses.

The hypotheses detailed in the manuscript follow directly from previous research on the evolved functions of pride and shame. Much empirical evidence exists supporting the theoretical perspective from which the authors derive their hypotheses; as such, these hypotheses are a natural extension from prior theory. Given that these hypotheses have already been tested and supported in multiple countries, it is highly plausible that they will be supported again in this nearly identical replication.

4. *Stage 1 Secondary Criterion #2*: The soundness of the methodology and analysis pipeline.

The methodology is sound and sensible, and the chosen analysis pipeline is sufficient for testing these hypotheses.

5. *Stage 1 Secondary Criterion #3*: Whether the authors have considered sufficient outcome-neutral conditions (e.g. absence of floor or ceiling effects; positive controls; other quality checks) for ensuring that the results obtained are able to test the stated hypotheses.

The authors included an attention check in Study 1 to ensure data quality. The participants evaluate a large number of different scenarios for each study, so they will provide a sufficiently wide range of responses that should preclude floor or ceiling effects. That said, providing a complete list of all the stimuli would bolster my confidence that each study contains scenarios that engender ratings of high, medium, and low levels of pride and shame. Additionally, using more items to assess anticipated pride/shame and social value may increase variability in responses if that is an issue.

Overall, I think the authors demonstrated a strong commitment to conducting a faithful replication of the original studies and to transparently planning their work and describing it to readers. My few suggestions are quite minor and none should be considered serious problems.

We thank both reviewers for their careful reading of the Stage 1 manuscript and their positive assessment.

Appendix C

March 25, 2020

Dr. Chris Chambers
Editor (Registered Reports and Replications), *Royal Society Open Science*

Dear Dr. Chambers,

Thank you for considering “Do pride and shame track the evaluative psychology of audiences?: Preregistered replications of Sznycer et al. (2016, 2017),” co-authored with Rie Chun and Dr. Daniel Sznycer, for publication in *Royal Society Open Science*. We are grateful for the IPA decision and have prepared the final manuscript for your further consideration. As requested, we have registered our approved protocol on the Open Science Framework as a Registered Report. It can be found here: <https://osf.io/8r9ah>. Thank you again for reviewing our work.

Sincerely,

Adam S. Cohen, Ph.D.
Department of Psychology
University of Hawai'i at Manoa
cohen9@hawaii.edu
<https://adamcohen3.github.io/>

Rie Chun
Department of Psychology
University of Hawai'i at Manoa
riechun@hawaii.edu

Daniel Sznycer
Department of Psychology
University of Montreal
daniel.sznycer@umontreal.ca

Appendix D

Response to Reviewers

We thank the editor and reviewers for reviewing the Stage 2 submission of our manuscript and appreciate the quick turnaround and positive feedback.

The only edit requested was to fix the scale of the figures. We thank Reviewer 1 for catching this typo. We have now fixed the figures.

We would also like to mention that we have updated the reference to Klein and colleagues (2018), which was previously under review but now is published.

–